# Graft-versus-Host Disease Modulation by Innate T Cells

**DOI:** 10.3390/ijms24044084

**Published:** 2023-02-17

**Authors:** Ying Fang, Yichen Zhu, Adam Kramer, Yuning Chen, Yan-Ruide Li, Lili Yang

**Affiliations:** 1Department of Microbiology, Immunology and Molecular Genetics, University of California, Los Angeles, CA 90095, USA; 2Eli and Edythe Broad Center of Regenerative Medicine and Stem Cell Research, University of California, Los Angeles, CA 90095, USA; 3Jonsson Comprehensive Cancer Center, David Geffen School of Medicine, University of California, Los Angeles, CA 90095, USA; 4Molecular Biology Institute, University of California, Los Angeles, CA 90095, USA

**Keywords:** graft-versus-host disease (GvHD), GvHD modulation, innate T cell, mucosal-associated invariant T (MAIT) cell, invariant natural killer T (iNKT) cell, gamma delta T (γδ T) cell, major histocompatibility complex (MHC), T-cell receptor (TCR)

## Abstract

Allogeneic cell therapies, defined by genetically mismatched transplantation, have the potential to become a cost-effective solution for cell-based cancer immunotherapy. However, this type of therapy is often accompanied by the development of graft-versus-host disease (GvHD), induced by the mismatched major histocompatibility complex (MHC) between healthy donors and recipients, leading to severe complications and death. To address this issue and increase the potential for allogeneic cell therapies in clinical practice, minimizing GvHD is a crucial challenge. Innate T cells, encompassing subsets of T lymphocytes including mucosal-associated invariant T (MAIT) cells, invariant natural killer T (iNKT) cells, and gamma delta T (γδ T) cells, offer a promising solution. These cells express MHC-independent T-cell receptors (TCRs), allowing them to avoid MHC recognition and thus GvHD. This review examines the biology of these three innate T-cell populations, evaluates research on their roles in GvHD modulation and allogeneic stem cell transplantation (allo HSCT), and explores the potential futures for these therapies.

## 1. Introduction

Graft-versus-host disease (GvHD) is a common complication associated with allogeneic transplantation such as allogeneic hematopoietic stem cell transplantation (allo HSCT) and chimeric antigen receptor T (CAR-T)-cell therapies. With their highly variable αβ T-cell receptors (TCRs), donor T cells bind with major histocompatibility complex (MHC) class I and II molecules widely expressed on recipient tissue cells [1]. When the TCR recognizes the MHC as foreign, donor T cells initiate the immune response and attack recipient cells, causing an alloreaction leading to GvHD. Moreover, in adoptive cell therapies without preconditioning, recipient T cells can also recognize mismatched donor MHC molecules and attack transplanted cells. However, even with MHC-matched donors such as siblings, patients develop acute and chronic GvHD in 25–40% and 40–60% of the cases, respectively [2]. Under these circumstances, GvHD is induced by the disparity between donor and recipient minor histocompatibility antigens (miHAs), peptides bound to MHC molecules that are capable of triggering a donor T cell immune response [3].

Mediators of GvHD include several populations of T cells, such as CD4^+^ T cells, CD8^+^ T cells, and CD4/CD8 double-positive populations. These T-cell populations induce GvHD via a perforin-dependent pathway [4] and secretion of interferons (IFNs) [5]. Different memory T-cell subsets have varying abilities to mediate GvHD based on their phenotypes. For instance, whereas CD62L^+^ naïve T cells and CD44^+^CD62L^+^ central memory T cells are capable of inducing GvHD due to their high alloreactivity [6,7], CD62L^-^ memory T cells do not cause GvHD in general [8,9]. In the case of Th17 cells, studies have indicated that GvHD progression is exacerbated after polarization [10] and that in vitro differentiation leads to severe GvHD with pulmonary damage [11]. Furthermore, CD3^+^CD4^+^CD25^hi^FoxP3^+^ regulatory T cells (Tregs) alleviate GvHD by directly regulating effector T cell function via the secretion of inhibitory cytokines [12].

Current therapies against GvHD work by manipulating mediators of the disease, the most common mediator being T cells. To address this, the most common treatment for GvHD is T-cell depletion (TCD), allowing accurate elimination of specific T-cell subsets, usually via magnetic-associated cell sorting [13]. Studies on CD34^+^ TCD in CAR-T-cell transplantation indicate that alloreactive T cells from the graft are effectively depleted and the incidence of GvHD and relapse is significantly decreased [14,15]. Manipulating other mediators of alloreactivity can also be useful, such as expanding Tregs to alleviate GvHD or recruiting NK and gamma delta T cells to restore immunity following depletion [16,17], considering that TCD may lead to diminished reactivity and efficacy of the grafted effector cells [13]. In addition, myeloid-derived suppressor cells (MDSCs), which are formed under chronic inflammation or infection and stimulated by signals including macrophage (M-)/granulocyte (G-)/granulocyte–macrophage (GM-) colony-stimulating factors (CSFs) and proinflammatory cytokines such as IFN-γ, IL-4, IL-6, and IL-23 [18,19], play a significant role in the treatment of GvHD with their immunosuppressive function, shielding grafted cells from alloreactivity. MDSCs are also capable of assisting Treg expansion and inhibiting the proliferation of T cells through cytokine secretion, suppressing the inflammation induced by GvHD [19].

One limitation associated with the depletion-based therapies for GvHD described above is that they focus on eradicating reactive T cells rather than addressing MHC recognition, severely limiting the efficacy of the initial adoptive cell transfer. A potential solution being explored is harnessing the MHC-independent properties of innate lymphocytes to limit GvHD responses in patients. Mucosal-associated invariant T (MAIT) cells, invariant natural killer T (iNKT) cells, and gamma delta T (γδ T) cells are three populations of innate T lymphocytes being explored for their potential in mediating GvHD responses (Table 1). With their semi-invariant chains and restricted number of α/β/γ chains [20], innate T cells induce cytotoxicity through MHC-independent mechanisms, enabling the transplantation of allogeneic effector cells without the risk of GvHD. In addition to their MHC-independent TCR activation, each of these innate T cells also has a unique mechanism to minimize GvHD (Figure 1). In this review, we elaborate on these innovative approaches to treat GvHD using innate T cell modulation.

## 2. MAIT Cell Modulation of GvHD

MAIT cells are a subset of innate T cells found in the blood, liver, and the epithelial layers of the lung and the respiratory and GI tracts [49,50]. These cells are quiescent until activated by microbial infections. MAIT cells exhibit a CD161^hi^ phenotype and have a semi-invariant TCR composed of an invariant α chain and a β chain from a variant β repertoire [26]. MAIT cells have several mechanisms for activation, including TCR-dependent and -independent pathways. The TCR-dependent pathway involves TCR recognition of riboflavin-derived antigens presented by MR1, an MHC class I-like molecule for MAIT cell activation [51]. The primary ligand for MR1 is 5-(2-oxopropylideneamino)-6-d-ribitylaminouracil (5-OP-RU), an intermediate of riboflavin biosynthesis by microbial pathogens [52]. Studies have demonstrated the effector activity of MAIT cells in vitro is significantly enhanced with the administration of synthetic 5-OP-RU, indicating its potential role in mediating MAIT populations [48]. The TCR-independent mechanism involves inflammatory cytokines including IL-7, IL-12, and IL 18 [50]. Upon activation, MAIT cells proliferate, accumulate in vivo, initiate cytotoxic function via perforin and granzymes, and secrete pro-inflammatory cytokines (e.g., IFN-γ, TNF-α, IL-15, IL-17, and GM-CSF) to recruit circulating effector cells in vivo [53].

The unique biology of MAIT cells allows them to minimize the occurrence of GvHD from MHC mismatch. MAIT cells have been shown to be crucial in both acute and chronic GvHD, particularly when it comes to complications of the gut and skin, where MAIT cells reside. In patients with allo HSCT, Gao et al. found that there was a higher risk of acute GvHD in grafts with lower frequencies of MAIT cells [21,25]. Acute GvHD may also be influenced by the intestinal microbiota: acute GvHD is more likely to develop when there are more non-riboflavin pathways [22], further providing evidence that MAIT cell activation plays a key role in GvHD modulation.

MAIT cells in allografts influence chronic GvHD as well. Although the long-term reconstitution of MAIT cells after allograft transplantation may be affected by various factors including the diversity of gut microbiota, different donor sources, and MAIT cell number/type in the transplanted tissue [24], the early reconstitution is influenced by MAIT cell proliferation after transplantation [26]. Higher MAIT cell counts in allografts reduce the risk of poor MAIT reconstitution and, consequently, the incidence of GvHD [24]. Along with clinical results, in vitro studies of CAR-MAIT cells showed that these cells exhibit high efficacy and safety against tumors, minimizing GvHD in allogeneic cell-based therapy [48]. The CAR-MAIT cell TCR identifies high levels of MR1 molecules on myeloid-cell-derived APCs, which have been found to exacerbate acute and chronic donor T-cell-induced GvHD; hence, CAR-MAIT cells may remove these myeloid APCs and diminish GvHD [48,54,55].

## 3. iNKT Cells Modulation of GvHD

iNKT cells are a subset of T cells that play a role in bridging the adaptive and innate immune systems with a variety of mechanisms [56,57]. As an innate lymphocyte population, iNKT cells express the semi-invariant Vα24Jα18 TCR in humans, paired with a limited Vβ chain [42,58,59,60]. Similar to other innate immune cells, iNKT cells exhibit less specificity and a quicker activation than adaptive lymphocytes [61]. iNKT cells recognize lipid antigens presented on CD1d [57,59], a non-polymorphic MHC class I-like molecule [42,57,59,61]. Due to the degree of conservation in the canonical TCR and CD1d molecules, interspecies cross-reactivity is possible [62]. For instance, mouse iNKT cells are capable of reacting with human CD1d molecules and vice versa, demonstrating iNKT cells’ potentially broad role in the immune system [62].

As mentioned before, one distinguishing characteristic of iNKT cells is that they recognize glycolipids presented on CD1d [60]. The majority of the known antigens are composed of a similar structure of a lipid tail buried into the CD1d surface protein and a sugar head group emerging to make contact with the iNKT TCR [62]. The glycolipid α-galactosylceramide (α-GalCer) was the first discovered crucial activator for iNKT cells [57,63]. Although iNKT cells possess both TCR α and β chains, evidence supports that recognition of CD1d is carried out via the TCRα chain with four essential amino acids: Asp94, Arg95, Gly96, and Ser97. The TCRβ chain is shown to not be involved in the binding process [64,65,66,67]. Although binding with cognate antigens, such as microbial glycolipids, can directly stimulate iNKT cells, indirect activation of iNKT cells occurs through two primary methods for pathogens that lack cognate antigens: partial CD1d-TCR-binding-dependent activation combined with antigen presenting cell (APC) stimulation or CD1d-independent activation [62].

iNKT cells have been shown to play a role in modulating the GvHD response in transplant patients [27,32,33,34,35,38,56,68]. With reduced intensity conditioning (RIC), total lymphoid irradiation (TLI), and antithymocyte globulin (ATG), iNKT cells are able to alleviate GvHD in transplant patients [27,41,69]. Moreover, host iNKT cells trigger the expansion of Tregs, which play an essential role in the immunosuppression required to avoid GvHD [31]. The severity of GvHD in humans has been found to be correlated with the persistence of iNKT. [29,70,71,72,73,74]. The incidence of GvHD is also reduced in grafts that include more donor iNKT cells [56]. For instance, one study has demonstrated that the number of iNKT cells in the cryopreserved graft significantly increased GvHD-free progression-free survival (GPFS) among patients undergoing peripheral blood stem cell transplantation (PBSCT) by an average of two years [28].

Because of iNKT cells’ potential in reducing GvHD, the therapeutic potential of iNKT cells is being increasingly studied. One method to enhance the therapeutic potential of iNKT cells is through ex vivo expansion via single antigenic stimulation [40]. In the study conducted by Trujjilo-Ocampo et. al., following enrichment from peripheral blood mononuclear cells (PBMCs), iNKT cells are cultured with antigen-presenting dendritic cells for two weeks with agonist glycolipids such as α-GalCer [40]. Expanded iNKT cells express high levels of CD4 and alleviate xenograft GvHD, as evidenced by a higher survival rate for the iNKT-treated mice, as well as significantly less severe GvHD features in the skin, small intestine, liver, and lung compared with those in the PBMC-only-treated mice. The suppression of the proliferation of conventional T cells was observed as well, which might be the consequence of strong TCR-mediated activation of responder T cells or the high ratio of responder T cells to iNKT cells [40].

Two main phenotypes of iNKTs in humans include CD4^+^ iNKT cells that secrete more IL-4 and CD4^-^CD94^+^ iNKT cells with a strengthened cytotoxic function [75]. These cells suppress GvHD by expansion manipulation through IL-4 secretion and peripheral tolerance pathways [33] and by controlling alloreactivity through CD4^-^ iNKT cells targeting recipient APCs [35]. Being supported by the biology of these two sublineages and the clinical evidence of the correlation between donor iNKT persistence and lower incidences of GvHD, preclinical studies have broadened the road for potential allogeneic therapies that utilize CD4^+^ or CD4^-^CD94^+^ iNKT cells as the source. A study has generated third-party hematopoietic stem cell (HSC)-engineered human iNKT (^3rd^HSC-iNKT) cells, which is accomplished through the combination of HSC gene engineering and in vitro HSC differentiation, demonstrating the ability of iNKT cells to simultaneously ameliorate GvHD while maintaining the anti-tumor response. Similar to the PBMC-derived endogenous CD4^-^ iNKT cells, ^3rd^HSC-iNKT cells secrete high levels of IFN-γ, TNF-α, granzyme B, and perforin, indicating the Th1 cytokine profile and cytotoxic potential. In this study, the iNKT cells’ suppression for GvHD is at least partially through the recognition of CD1d. Such an advantage in GvHD reduction is also attributed to the Treg expansion induced by IL-4 as mentioned above. One interesting point about this study is that although the mechanisms vary, CD4^+^ iNKT cells are also involved in GvHD amelioration. By controlling resident immune cells and secreting regulatory cytokines, iNKT cells are found to be potent mediators of GvHD in allograft patients, and engineering mechanisms to harness this potential is being investigated for its therapeutic benefit [39].

## 4. γδ T Cell Modulation of GvHD

γδ T cells are a highly heterogeneous population of T lymphocytes that exhibit qualities of both innate and adaptive immune cells. They are defined by a γδ TCR unique from the conventional αβ T cells in their antigen recognition, anatomical distribution, and killing mechanisms [76]. γδ T cells normally constitute between 1–10% of the total T cells in the human body and operate independently of HLA recognition during the initial phases of the immune response [43]. The γδ TCR loci are the T-cell receptor gamma (TRG) and the T-cell receptor delta (TRD) loci, and rearrangement is dependent on recombination activating genes (RAG). These loci are much more restricted than in αβ T cells, with only six functional TRG V segment genes and eight functional TRD genes (compared with 70 *Vα* and 52 *Vβ* genes) [77]. The TCR composition largely determines the localization of the T cell, with Vδ1, Vδ3, or Vδ5 TCRs localizing in epithelial tissues and Vγ9Vδ2 more commonly circulating in the peripheral blood [78,79]. The vast majority of γδ T cells do not express CD4 or CD8 coreceptors but rather express several NK receptors for NK-like function (e.g., CD16, NKG2D, MIC-A, MIC-B, and UL16) [80,81].

Whereas αβ T cells migrate to the thymus, many γδ T migrate to tissues such as the epidermis, dermis, intestines, or uterus. In addition, unlike conventional αβ T cells, γδ T cells recognize molecules independently of MHC [76]. TCR composition and coreceptor expression largely dictate the physiological distribution and antigen specificity of different γδ T cell subtypes. The most common peripherally circulating γδ T cells, Vγ9Vδ2 γδ T cells, recognize phosphoantigens produced by microbes or malignancies, and tissue-localized Vδ1 and Vδ3 γδ T cells recognize molecules presented by the CD1 family of surface receptors [77]. Vγ8Vδ3 peripheral blood γδ T cells recognize annexin A2, and Vγ4Vδ5 γδ T cells recognize endothelial protein C receptors in the peripheral blood [82,83].

Effector function in γδ T cells is activated through various means, including secreting regulatory cytokines, releasing perforin, granzymes, and IFN-γ, and antibody-dependent cellular cytotoxicity (ADCC) via CD16 [84]. Effector function in γδ T cells may vary depending on the individual cell’s niche, with intestinal-resident γδ T cells secreting keratinocyte growth factor for epithelial homeostasis [77]. Other effector functions may be induced either dependent or independent on TCR recognition. For example, IL-1β- and IL-23- induced cytokine production allows γδ T cells to secrete IL-17 and other regulatory cytokines in humans [85]. Furthermore, TCR- and NK-marker activation may enable γδ T cells to enter a pro-inflammatory state, secrete IFN-γ, TNF-α, and IL-17, and induce cells to enter a state of antigen presentation, thereby promoting the adaptive immune response in CD4^+^ and CD8^+^ T cells [10]. In this manner, γδ T cells serve a role in bridging the innate and adaptive immune responses, activating with and without antigen stimulation.

The nature of γδ T cells’ MHC-independent activation suggests that they may play a vital role in mediating GvHD for allogeneic cell therapies. In patients with allogeneic stem cell transplantation for hematological malignancies, higher concentrations of γδ T cells about two months after transplantation correlated with improved overall survival and relapse-free survival. Moreover, the risk of acute GvHD one month after transplantation was greatly reduced for patients with higher levels of γδ T cells [43]. Speculatively, allogeneic γδ T cells may alleviate GvHD by immunoregulatory actions such as IL-4 secretion or cytotoxic effects on CD277-expressing APCs [48,86]. These data suggest a role for these lymphocytes in protecting against tumor cells and against alloreactivity.

In addition to being associated with better prognosis in stem cell transplant patients, γδ T cells show great potential as an allogeneic cell product for cancer immunotherapy owing to their absence of MHC restriction, inherent antitumor abilities, and ability to act as an APC [87]. Several methods have been studied to utilize γδ T cells as an adoptive cell product. Most commonly aminobisphosphonates, known stimulants of γδ T cells, have been utilized for ex vivo expansion and activation. Gertner-Dardenne et al. demonstrated expanded γδ T lymphocytes were capable of efficient killing of AML blasts via TCR- and DNAM-1-mediated perforin/granzyme activation both in vitro and in murine models [88]. Further studies with anti-BTN3A 20.1 monoclonal antibodies have demonstrated in vitro expansion and Vγ9Vδ2-T-cell-mediated killing of AML in vitro. The monoclonal antibody activation of Vγ9Vδ2 T lymphocytes was demonstrated to be instigated via both TCR and BTN3A pathways, with the anti-BTN3A 20.1 antibodies enabling effector cells to target cells resistant to typical Vγ9Vδ2-T-cell-mediated lysis [89]. Similarly, FDA-approved monoclonal antibodies (i.e., rituximab and alemtuzumab) and the bispecific T-cell engager (BiTE) blinatumomab have shown improved survival against hematological malignancies in murine models [87]. Monoclonal antibodies targeting hematological malignancies activate γδ T cells via CD16-mediated ADCC function, and BiTE molecules activate via CD3 stimulation and dual engagement of tumor and T cells.

For adoptive cell transfer, CAR-γδ T cells show promise as an allogeneic therapy due to MHC-independent TCR recognition. Data is limited on the success of CAR-γδ T cells; however, preclinical data demonstrates enhanced killing against leukemia, targeting both CD19^+^ and CD19^-^ tumor cells in murine models [14]. Adoptive transfer has also been tested in preclinical trials for CD20-directed CAR-γδ T cells and showed promise against B-cell lymphoma, serving as the basis for a phase I clinical trial for patients with CD20-positive B-cell malignancies (NCT04735471) [90]. Preclinical success with CAR-γδ T cells supports a multipronged activation resulting from NK activation markers and CAR-specified activation of the engineered cells.

Allogeneic cell therapies based on γδ T cells show promise across several pathways of administration. The positive association of γδ T cell level and patient outcome and prognosis in regard to tumor relapse and occurrence of GvHD heavily indicates that γδ T cells play a role in the modulation of alloreactivity and tumor persistence. Because of this, several means of utilizing the MHC-independence and high tumor reactivity of γδ T cells are being explored for allogeneic cell therapies. However, other research has suggested that γδ T cells may have a negative impact in GvHD. Wu et al. discovered that γδ T cells enhanced CD4 T-cell migration via the SDF-1–CXCR4 axis, exacerbating acute GvHD post allo HSCT [91]. As a result, more research into the influence and activities of γδ T cells in modulating GvHD is warranted.

## 5. Discussion

Reducing the occurrence of GvHD events in transplant patients has the potential to broaden the population able to receive newly developed adoptive transfer treatments. Due to their MHC-independent targeting mechanism, innate-like T-cell subsets, which are distinguished by their semi-invariant TCRs, show promise to reduce the occurrence of GvHD. In this review, we comprehensively summarized the biological functions of MAIT, iNKT, and γδ T cells and their role in the modulation of GvHD responses. Innate T cells, as previously noted, do not recognize MHC molecules, reducing GvHD by limiting the detection of foreign antigens by surveilling immune cells. Low frequencies of MAIT, iNKT, and γδ T cells are correlated with an elevated risk of GvHD and a decreased overall survival rate in patients following allogeneic stem cell transplantation, suggesting innate T cells play a role in the modulation of GvHD in recipients of allogeneic transplants [28,43,92]. Therefore, these T-cell subtypes are receiving increasing attention to address GvHD in transplant patients.

Although there is a correlation between the presence of innate T cells and the risk of GvHD, the mechanisms of how GvHD is modulated remain unclear for MAIT cells and γδ T cells. However, for iNKT cells, GvHD-mediation is better studied. For instance, we understand that iNKT cells prevent and reverse chronic GvHD in murine models by expanding donor Tregs via cytokine stimulation [38] and CD4^+^ iNKT cells were found to protect mice from fatal GvHD [33]. Furthermore, human CD4^−^ iNKT cells were reported to suppress the numbers and maturation levels of recipient APCs (mostly dendritic cells) during an allogeneic immune response [35]. Schimid et al. proposed that iNKT cells could induce preferential apoptosis of circulating conventional dendritic cells (cDCs), key stimulators of the alloreactive T cell response [36]. This selective apoptosis could result in a relative expansion of beneficial plasmacytoid dendritic cells (pDCs) and a decrease in the activation and proliferation of T cells from healthy donors and GvHD patients [36]. Although there is overlap in the mechanisms of each, the overall pathway for the suppression of alloreactions in human iNKT cells remains undetermined [36,56].

Despite little understanding of the mechanisms of innate T cells in GvHD modulation, there are four primary approaches that these cells could be promisingly incorporated into for allogeneic transplantation (Figure 2). First, innate T cells have a great safety profile and could be employed as the primary effector against cancer and other disorders through their NK-like and TCR-mediated cytotoxic functions [93]. Our lab developed a stem cell platform to generate off-the-shelf allogeneic iNKT cells for cancer [37]. These cells persist with a strong capacity for tumor destruction together with minimal evidence of GvHD in vitro and a long-lasting effect in mice. γδ T cells and MAIT cells have also been discovered to display similar powerful antitumor properties [48,93]. Second, HSCs could be engineered in vitro with genes for specific invariant TCRs to generate TCR-engineered allo HSCT [94]. The engineered HSCs will continuously produce specific T-cell subsets post-HSC reconstitution, thus providing patients with a long-term supply of therapeutic cells and reducing the risk of GvHD for the duration of treatment [94]. Thirdly, to create a more balanced population of infused cells in allo HSCT, adoptive transfer of donor innate T cells could be used in low doses to protect from GvHD [34]. Lastly, allo HSCT might be administered with third-party innate T cells to reduce GvHD while maintaining the antileukemia efficacy [39]. For example, allogeneic iNKT cells were found to target myeloid-derived antigen presenting cells and could be administered as an off-the-shelf therapy in combination with allo HSCT [33,39].

Innate T cell-related therapy shows potential over traditional methods of GvHD reduction. Current methods try to improve host immunoregulation, such as by increasing Treg concentrations or by chemokine/cytokine modulation, or to decrease transplant potency by T-cell depletion [95,96]. These techniques reduce treatment effectiveness and increase safety concerns. Instead, by producing ideal innate-like cell products prior to infusion, therapies may involve fewer complications while reducing the risk of GvHD reactions. The indirect modulation of GvHD exhibited by these cells also supports the potential to rely less on recipient preconditioning as the infused innate T cells suppress reactive T lymphocytes. Furthermore, with a far-reduced risk of GvHD, genetic engineering approaches could be employed to further improve safety and efficacy. For instance, suicide genes could be engineered in innate T cells to eradicate effector cells once remission has been achieved and reduce risk of other toxicities related to the grafted cells [97]. Incorporating these safety steps into the development process for adoptive cell transfers ensures a built-in safety mechanism in the result of unforeseen reactions not possible in conventional MHC-restricted therapies.

However, innate T cells do hold limitations. Since innate T cells compile a small fraction of human PBMCs, cell isolation and expansion require a significant amount of time, inhibiting higher yields and large-scale manufacturing [37,98]. This challenge may be addressed by utilizing stem cells, which produce up to 10,000 doses of therapeutic cells from a single donor [39,99]. Additionally, as was previously mentioned, the development of these therapies may be affected by unforeseen reactions from the transplantation of allogeneic innate-like T cells, likely due to undiscovered immune pathways in these cells. Therefore, additional research is required to safely and efficiently engineer innate T cells for the prevention or treatment of GvHD.

## Figures and Tables

**Figure 1 ijms-24-04084-f001:**
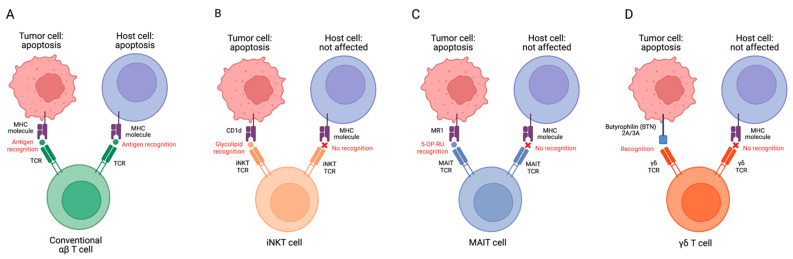
Targeting tumor and host cells by conventional αβ T (**A**), iNKT (**B**), MAIT (**C**), and γδ T cells (**D**). MHC–antigen–TCR interactions allow T cells to recognize target cells. MHC molecules on host cells could be recognized by conventional T cell TCRs after allogeneic cell infusion, resulting in GvHD. However, innate T cell TCRs do not recognize mismatched MHCs and protein antigens, therefore, these cells have no GVHD risk. MR1, major histocompatibility complex, class I-related protein; 5-OP-RU, 5-(2-oxopropylideneamino)-6-D-ribitylaminouracil.

**Figure 2 ijms-24-04084-f002:**
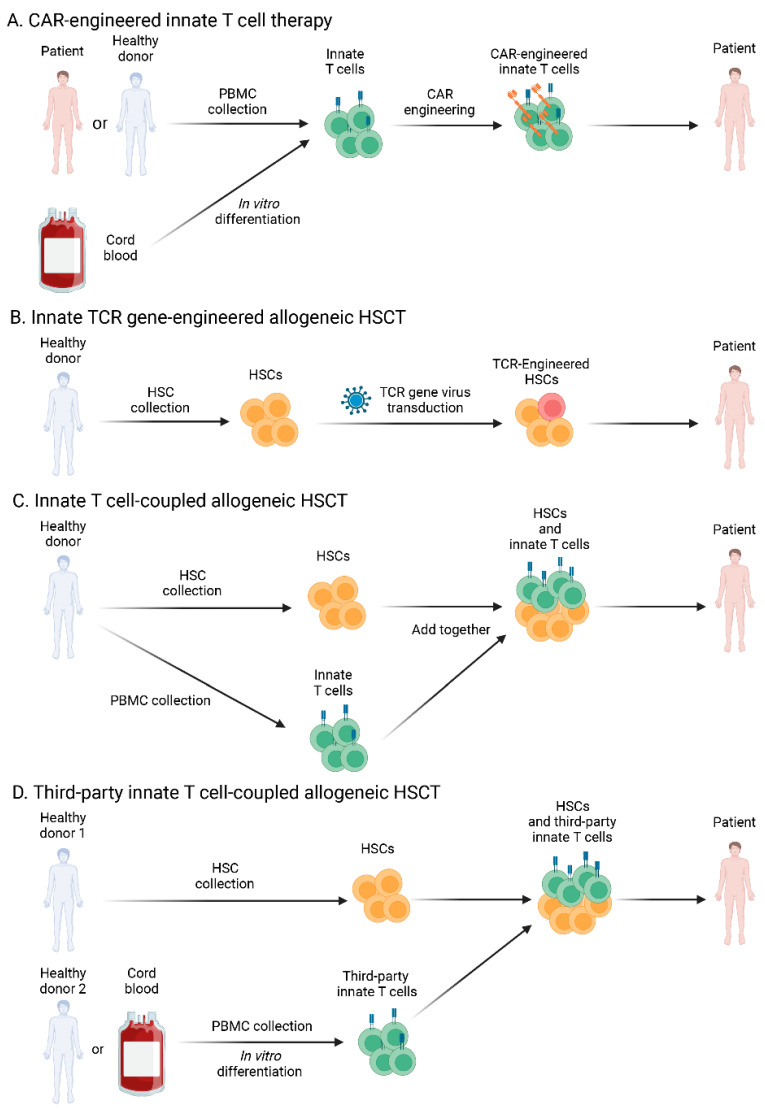
GvHD modulation of innate T cells in cancer immunotherapy. (**A**) CAR-engineered innate T cells are generated from human peripheral blood mononuclear cells (PBMCs) or cord blood hematopoietic stem cells (HSCs). These CAR-engineered innate T cells could target cancers with high efficacy and low GvHD risk. (**B**) Allogeneic HSCs are engineered with innate T cell TCRs via lentiviral or retroviral transduction. Post-transplant reconstitution, innate T cells could be produced continuously to target cancers with reduced GvHD. (**C**) Donor innate T cells are expanded and coupled with HSCs for allogeneic HSC transplantation (HSCT). A low dose of innate T cells could potentially protect patients from GvHD. (**D**) Third-party innate T cells are generated from healthy donor PBMCs or cord blood HSCs and infused to cancer patients with allogeneic HSCs. Off-the-shelf innate T cells could provide a more convenient strategy for GvHD reduction.

**Table 1 ijms-24-04084-t001:** GvHD modulation by three innate T cells, including MAIT, iNKT, and γδ T cells.

Innate T Cell Types	Clinical or Preclinical	GvHD Modulation
MAIT	Clinical	An increased number of MAIT cells is associated with improved overall survival and less GvHD [21,22,23,24,25].
	Preclinical	MAIT cells suppressed conventional CD4+ T-cell proliferation [25]. MAIT cells regulate GvHD in part by the generation of IL17A [26].
iNKT	Clinical	A higher number of iNKT cells is correlated with better GvHD-free survival [27,28,29,30].
	Preclinical	Host iNKT cells trigger the expansion of donor Tregs through an interleukin-4-dependent pathway [31]. CD4^+^ iNKT cells protect mice from GvHD [32,33,34]. CD4^−^ iNKT cell subset controls GvHD better than its CD4^+^ counterpart by inducing apoptosis of dendritic cells [35]. iNKT cells induced selective apoptosis of conventional dendritic cells but not beneficial plasmacytoid dendritic cells [36]. Allogeneic iNKT cells could target tumor cells without GvHD risk [37]. Donor iNKT cells could prevent and reverse chronic GvHD in murine models [38]. Host/allogeneic iNKT cells could ameliorate GvHD while preserving antitumor effects [32,39,40,41]. iNKT2 and iNKT17 cells are responsible for the immune regulatory properties instead of iNKT1 cells [42].
γδ T	Clinical	An increased number of γδ T cells is associated with less acute GvHD [22,43]. Patients receiving higher concentrations of donor γδ T cells have a more frequent incidence of acute GvHD [44,45]. The γδ Treg proportion has a negative correlation with acute GvHD incidence [46].
	Preclinical	G-CSF-induced γδ T cells suppress CD4^+^ T-cell proliferation and regulates acute GvHD [47]. CAR-engineered gdT cells lead to a lower incidence of GvH response [48].

## Data Availability

Not applicable.

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
