# Peer review of "Graft-versus-Host Disease Modulation by Innate T Cells"

_ijms, 2023, doi:10.3390/ijms24044084_

Round 1

Reviewer 1 Report

Great paper, however please note the following points;

1. There are so many complicated terminologies and receptor biology involved that it can be misleading or confusing. The authors may create figures for better understanding. 

2. Can GvHD be correlated from rodent model to human? I believe it would be best to focus on only either preclinical or clinical.

3. There are alot of cases where the author would give an example out of nowhere and it sometimes throughs the paragraph out of context or lead the reader to confusion. Please include more transition between paragraphs. 

4. There are some citations listed from the late 1990s and the early 2000s. It would be recommended to cite recent word. 

Author Response

1. There are so many complicated terminologies and receptor biology involved that it can be misleading or confusing. The authors may create figures for better understanding. 

Response: We appreciate the reviewer’s feedback. We added a figure in our manuscript to illustrate different activation mechanisms (via MHC-dependent or MHC-independent pathways) of MAIT, iNKT, γδ T, and conventional T cells. We also include specific ligands/antigens/receptors involved in these pathways. See Page 4-5, Figure 1: Targeting tumor and host cells by conventional αβ T, iNKT, MAIT, and γδ T cells.

2. Can GvHD be correlated from rodent model to human? I believe it would be best to focus on only either preclinical or clinical.

Response: We appreciate the review’s comment and acknowledge the reviewer’s concern. As we elucidated, using innate T cells to minimize GvHD is a cutting-edge approach. In present, there is limited clinical data available that directly utilizes innate T cells as a cell source to reduce GvHD. Instead, the available clinical data only suggest a correlation between increased innate T cell population in allograft and reduced severity of GvHD. Based on these correlations, further studies have been conducted to determine if innate T cells can be a safe and GvHD-free cell source for allo HSCT or allo CAR T therapy. These validation tests cannot be performed on patients due to ethical considerations. Therefore, current research on the use of innate T cells as a means of minimizing GvHD is limited to studies conducted on rodent models. The limitations of the rodent model were various, including highly distinct immunoreaction in murine model and human after allotransplantation, as well as various patient-specific factors (reference: https://doi.org/10.1016/j.isci.2022.104859).

3. There are a lot of cases where the author would give an example out of nowhere and it sometimes throughs the paragraph out of context or lead the reader to confusion. Please include more transition between paragraphs. 

Response: We thank the reviewer for the constructive suggestions. We rearranged the order of several examples and use more transitions in the manuscript to make the flow more logically fluent.

We edited these paragraphs in Page 5 and 6:

“iNKT cells have been shown to play a role in modulating GvHD response in transplant patients [19,30-33,36,52,64]. With reduced intensity conditioning (RIC), total lymphoid irradiation (TLI), and antithymocyte globulin (ATG), iNKT cells are able to alleviate GvHD in transplant patients [39,65]. Moreover, host iNKT cells trigger the expansion of Tregs, which play an essential role in the immunosuppression required to avoid GvHD [29]. The severity of GvHD in humans has been found to be correlated with the persistence of iNKT. [21,67-71]. The incidence of GvHD is also reduced in grafts that include more donor iNKT cells [52]. For instance, one study has demonstrated that the number of iNKT cells in the cryopreserved graft significantly increased GvHD-free progression-free survival (GPFS) among patients undergoing peripheral blood stem cell transplantation (PBSCT) by an average of two years [20].”

“Because of iNKT cells’ potential in reducing GvHD, the therapeutic potential of iNKT cells is being increasingly studied. One method to enhance the therapeutic potential of iNKT cells is through ex vivoexpansion via single antigenic stimulation [38]. In the study conducted by Trujjilo-Ocampo et. al., following enrichment from peripheral blood mononuclear cells (PBMCs), iNKT cells are cultured with antigen-presenting dendritic cells for two weeks with agonist glycolipids such as α-GalCer [38]. Expanded iNKT cells express high levels of CD4 and alleviate xenograft GvHD, as evidenced by a higher survival rate of the iNKT-treated mice, as well as significantly less severe GvHD features in skin, small intestine, liver, and lung compared to those in the PBMC only-treated mice. The suppression of the proliferation of conventional T cells was observed as well, which might be the consequence of strong TCR-mediated activation of responder T cells or the high ratio of responder T cells to iNKT cells [38].”

4. There are some citations listed from the late 1990s and the early 2000s. It would be recommended to cite recent word.

Response: We appreciate the reviewer’s comments. We updated our citation lists with more recent work (published after 2018).

Reviewer 2 Report

Dear Authors,

In my opinion, you did great work on this review.

I have only minor issues listed below:

Either in the manuscript or in table 1 please change the order accordingly: iNKT, MAIT, and γδ T cells or MAIT, iNKT, and γδ T cells.

This makes the reader easy to follow.

Lane 289

Nevertheless, although there is a correlation between the presence of innate T cells and the risk of GvHD, the mechanisms of how GvHD is modulated remains unclear for 289

The sentence is not connected grammatically, it starts like a new paragraph in lane 301. Please correct it.

Besides, I don’t see any reference for Figure 1 in the text. I think it should be somewhere in the paragraph starting from lane 314. 

Author Response

In my opinion, you did great work on this review.

I have only minor issues listed below:

Either in the manuscript or in table 1 please change the order accordingly: iNKT, MAIT, and γδ T cells or MAIT, iNKT, and γδ T cells.

This makes the reader easy to follow.

Response: We thank the reviewer for the comments. In Table 1, we changed the order into: MAIT, iNKT, and γδ T cells. See Page 3-4, Table 1. GvHD modulation by three innate T cells, including iNKT, MAIT, and γδ T cells.

Lane 289

Nevertheless, although there is a correlation between the presence of innate T cells and the risk of GvHD, the mechanisms of how GvHD is modulated remains unclear for 289

The sentence is not connected grammatically, it starts like a new paragraph in lane 301. Please correct it.

Response: We appreciate the reviewer’s feedback. In Page 8, we changed Line 299 into:

“Nevertheless, although there is a correlation between the presence of innate T cells and the risk of GvHD, the mechanisms of how GvHD is modulated remain unclear forMAIT cells and γδ T cells.”

Besides, I don’t see any reference for Figure 1 in the text. I think it should be somewhere in the paragraph starting from lane 314. 

Response: We appreciate the reviewer’s comments. In Page 10, Line 323-325 of the manuscript, we reference Figure 2 (Because we added one figure in previous paragraph, this Figure 1 now becomes Figure 2) in this following sentence:

“Despite little understanding of the mechanisms of innate T cells in GvHD modulation, there are four primary approaches that these cells could be promisingly incorporated into for allogeneic transplantation (Figure 1).”

Reviewer 3 Report

This manuscript by Fang et al. reviews the benefits of innate T-cell populations (MAIT, iNKT and gamma-delta) in reducing graft versus host disease (GvHD) and how these populations might be exploited for allo HSCT. This manuscript is well-written and will be a useful resource to the GvHD field. The only drawback is that the authors do not speculate on how MAIT and gamma-delta populations might reduce GvHD. Are gamma-delta cells important for appropriate immune reconstitution in HSCT patients? Another minor critique is that the authors have not alluded to any studies that show possible deleterious effects of the cell populations; for example, Wu et al.  Front. Immunol., 14 July 2021. The only other comments relate to a few references:

Is ref. 75 on line 209 correct for the localization of the different gamma-delta variants?

Is ref. 77 on line 220 correct for the V-gamma8-V-delta3 recognition of annexin A2?

Author Response

This manuscript by Fang et al. reviews the benefits of innate T-cell populations (MAIT, iNKT and gamma-delta) in reducing graft versus host disease (GvHD) and how these populations might be exploited for allo HSCT. This manuscript is well-written and will be a useful resource to the GvHD field. The only drawback is that the authors do not speculate on how MAIT and gamma-delta populations might reduce GvHD. Are gamma-delta cells important for appropriate immune reconstitution in HSCT patients? Another minor critique is that the authors have not alluded to any studies that show possible deleterious effects of the cell populations; for example, Wu et al.  Front. Immunol., 14 July 2021. The only other comments relate to a few references:

Response: We appreciate the reviewer’s comments. To speculate on how MAIT and gamma-delta populations might reduce GvHD, In page 5, we added “CAR-MAIT cell TCR identifies high levels of MR1 molecules on myeloid cell-derived APCs, which have been found to exacerbate acute and chronic donor T cell-induced GvHD; hence, CAR-MAIT cells may remove these myeloid APCs and diminish GvHD.”

In page 7, we added “Speculatively, allogeneic γδ T cells may alleviate GvHD by immunoregulatory actions such as IL-4 secretion or cytotoxic effects on CD277-expressing APCs”

In page 7, we highlighted “In patients with allogeneic stem cells transplantation for hematological malignancies, higher concentrations of γδ T cells about two months after transplantation correlated with improved overall survival and relapse-free survival.”

Additionally, we discuss the related studies that show possible deleterious effects of γδ T cells: “However, other researches have suggested that γδ T cells may have a negative impact in GvHD. Wu et al. discovered that γδ T cells enhanced CD4 T cell migration via the SDF-1-CXCR4 axis, exacerbating acute GvHD post alloHSCT. As a result, more researches into the influence and activities of γδ T cells in modulating GvHD is warranted.” We also included the related information in Table 1.

Is ref. 75 on line 209 correct for the localization of the different gamma-delta variants?

Thanks. We updated the new reference.

Is ref. 77 on line 220 correct for the V-gamma8-V-delta3 recognition of annexin A2?

Thanks. We updated the new reference.

Reviewer 4 Report

Review for the paper “Graft-Versus-Host Disease Modulation by Innate T Cell”

 The paper entitled “Graft-Versus-Host Disease Modulation by Innate T Cell” reviewed the literature regarding the innovative approaches to treat GvHD using innate T cell modulation.

There are three innate T cells, including iNKT, MAIT, and γδ T cell with a role in GvHD. The authors summarized the biological functions of MAIT, iNKT, and γδ T cells and their role in the modulation of GvHD responses.

The results of the literature suggested innate T cells play a role in the modulation of GvHD in recipients of allogeneic transplants.

Innate-like T cell subsets show promise to reduce the occurrence of GvHD.

The results are promising for the prevention or treatment of GvHD.

The presentation is well organized, the writing style is clear. The references are adequate. 

Author Response

Response: We thank for the reviewer’s positive response. We would like to express our gratitude for your time and expertise in reviewing our manuscript. Your detailed comments and support are greatly appreciated.

Round 2

Reviewer 1 Report

None. Great improvement.